# Group privacy for personalized federated learning

**Filippo Galli**
Scuola Normale Superiore
Pisa, Italy
filippo.galli@sns.it

**Sayan Biswas**
INRIA, LIX, École Polytechnique
Palaiseau, France
sayan.biswas@inria.fr

**Kangsoo Jung**
INRIA
Palaiseau, France
gangsoo.zeong@inria.fr

**Tommaso Cucinotta**
Scuola Superiore Sant'Anna
Pisa, Italy
tommaso.cucinotta@santannapisa.it

**Catuscia Palamidessi**
INRIA, LIX, École Polytechnique
Palaiseau, France
catuscia@lix.polytechnique.fr

## Abstract

Federated learning exposes the participating clients to issues of leakage of private information from the client-server communication and the lack of personalization of the global model. To address both the problems, we investigate the use of metric-based local privacy mechanisms and model personalization. These are based on operations performed directly in the parameter space, i.e. sanitization of the model parameters by the clients and clustering of model parameters by the server.

## 1 Introduction

With the modern developments in machine learning, user data collection has become ubiquitous, often disclosing sensitive personal information with increasing risks of users' privacy violations [20, 26]. To try and curb such threats, Federated Learning (FL) [23] was introduced as a collaborative machine learning paradigm where the users' devices, on top of harvesting user data, directly train a global predictive model, without ever sending the raw data to a central server. On the one hand, this paradigm has received much attention with the appealing promises of guaranteeing user privacy and model performance. On the other hand, given the heterogeneity of the data distributions among clients, training convergence is not guaranteed and model utility may be reduced by local updates. Many works have thus focused on the topic of personalized federated learning, to tailor a set of models to clusters of users with similar data distributions [14, 22, 27]. On a similar note, other lines of work have also showed that relying on avoiding the release of user's raw data only provides a lax protection to potential attacks violating the users' privacy [16], [25], [31]. To tackle this problem, researchers have been exploring the application of Differential Privacy (DP) [11, 12] to federated learning, in order to quantify and provide privacy to users participating in the optimization. The goal of differential privacy mechanisms is to introduce randomness in the information released by the clients, such that each user's contribution to the final model can be made probabilistically indistinguishable up to a certain likelihood factor. To bound this factor, the domain of secrets (i.e. the parameter space in FL) is artificially bounded, be it to provide central [5, 24] or local DP guarantees [28, 30]. When users share with the central server their locally updated models for averaging,

Workshop on Federated Learning: Recent Advances and New Challenges, in Conjunction with NeurIPS 2022 (FL-NeurIPS'22). This workshop does not have official proceedings and this paper is non-archival.

constraining the optimization to a subset of $\mathbb{R}^n$ can have destructive effects, e.g. when the optimal model parameters for a certain cluster of users may be found outside such bounded domain. Therefore, in this work we aim to address the problems of personalization and local privacy protection by adopting a generalization of DP, i.e. $d$-privacy or metric-based privacy [9]. This notion of privacy does not require a bounded domain and provides guarantees dependent on the distance between any two points in the parameter space. Thus, under the minor assumption that clients with similar data distributions will have similar optimal fitting parameters, $d$-privacy will provide them with stronger indistinguishability guarantees. Conversely, privacy guarantees degrade gracefully for clients whose data distributions are vastly different.

## 2 Background

**Related works**  Federated optimization has shown to be under-performing when the local datasets are samples of non-congruent distributions, failing to minimize both the local and global objectives at the same time. In [14, 22, 27], the authors investigate different meta-algorithms for personalization. Claims of user privacy preservation are based solely on the clients releasing updated models (or model updates) instead of transferring the raw data to the server, with potentially dramatic effects. To confront this issue, a number of works have focused on the privatization of the (federated) optimization algorithm under the framework of DP [2,5,13,24] who adopt DP to provide defenses against an *honest-but-curious* adversary. Even in this setting though, no protection is guaranteed against sample reconstruction from the local datasets [31], using the client updates. Different strategies have been tried to provide local privacy guarantees, either from the perspective of cryptography [7], or under the framework of local DP [3,17,28]. In particular in [17] the authors address the problem of personalized and locally differentially private federated learning, but for the simple case of convex, 1-Lipschitz cost functions of the inputs. Note that this assumption is unrealistic in most machine learning models, and it excludes many statistical modeling techniques, notably neural networks.

**Personalized federated learning**  The problem can be cast under the framework of stochastic optimization and we adopt the notation of [14] to find the set of minimizers $\theta_j^* \in \mathbb{R}^n$ with $j \in \{1, \ldots, k\}$ of the cost functions

$$F(\theta_j) = \mathbb{E}_{z \sim \mathcal{D}_j}\left[f(\theta_j; z)\right], \tag{1}$$

where $\{\mathcal{D}_1, \ldots, \mathcal{D}_k\}$ are the data distributions which can only be accessed through a collection of client datasets $Z_c = \{z | z \sim \mathcal{D}_j, z \in \mathbb{D}\}$ for some $j \in \{1, \ldots, k\}$ with $c \in C = \{1, \ldots, N\}$ the set of clients, and $\mathbb{D}$ a generic domain of data points. $C$ is partitioned in $k$ disjoint sets

$$S_j^* = \{c \in C \mid \forall z \in Z_c, \ z \sim \mathcal{D}_j\} \quad \forall j \in \{1, \ldots, k\} \tag{2}$$

The mapping $c \to j$ is unknown and we rely on estimates $S_j$ of the membership of $Z_c$ to compute the empirical cost functions

$$\tilde{F}(\theta_j) = \frac{1}{|S_j|} \sum_{c \in S_j} \tilde{F}_c(\theta_j; Z_c); \qquad \tilde{F}_c(\theta_j; Z_c) = \frac{1}{|Z_c|} \sum_{z_i \in Z_c} f(\theta; z_i) \tag{3}$$

The cost function $f \colon \mathbb{R}^n \times \mathbb{D} \mapsto \mathbb{R}_{\geq 0}$ is applied on $z \in \mathbb{D}$, parametrized by the vector $\theta_j \in \mathbb{R}^n$. Thus, the optimization aims to find, $\forall j \in \{1, \ldots, k\}$,

$$\tilde{\theta}_j^* = \underset{\theta_j}{\arg \min} \ \tilde{F}(\theta_j) \tag{4}$$

**Privacy**  $d$-privacy [9] is a generalization of DP for any domain $\mathcal{X}$, representing the space of original data, endowed with a distance measure $d \colon \mathcal{X}^2 \mapsto \mathbb{R}_{\geq 0}$, and any space of secrets $\mathcal{Y}$. A random mechanism $\mathcal{R} : \mathcal{X} \mapsto \mathcal{Y}$ is called $\varepsilon$-$d$-private if for all $x_1, x_2 \in \mathcal{X}$ and measurable $S \subseteq \mathcal{Y}$:

$$\mathbb{P}\left[\mathcal{R}(x_1) \in S\right] \leq e^{\varepsilon d(x_1, x_2)} \mathbb{P}\left[\mathcal{R}(x_2) \in S\right] \tag{5}$$

Note that when $\mathcal{X}$ is the domain of databases, and $d$ is the distance on the Hamming graph of their adjacency relation, then Equation (5) results in the standard definition of DP in [11,12]. In this work we will have though that $\theta \in \mathbb{R}^n = \mathcal{X} = \mathcal{Y}$. The main motivation behind the use of $d$-privacy is to preserve the topology of the parameter distributions among clients, i.e. to have that, in expectation, clients with close model parameters in the non-privatized space $\mathcal{X}$ will communicate close model parameters in the privatized space $\mathcal{Y}$.

# 3 An algorithm for private and personalized federated learning

We propose an algorithm for personalized federated learning with local guarantees to provide group privacy (Algorithm 1). Locality refers to the sanitization of the information released by the client to the server, whereas group privacy refers to indistinguishability with respect to a neighborhood of clients, defined with respect to a certain distance metric. Thus we proceed to define *neighborhood* and *group*.

**Definition 3.1.** For any model parametrized by $\theta \in \mathbb{R}^n$, we define its $r$-*neighborhood* as the set of points in the parameter space which are at a $L_2$ distance of at most $r$ from $\theta$, i.e., $\{x \in \mathbb{R}^n \colon \|\theta, x\|_2 \leq r\}$. Clients whose models are parametrized by $\theta \in \mathbb{R}^n$ in the same $r$-neighborhood are said to be in the same *group*, or *cluster*.

Algorithm 1 is motivated by the Iterative Federated Clustering Algorithm (IFCA) [14] and builds on top of it to provide formal privacy guarantees. The main differences lie in the introduction of the `SanitizeUpdate` function described in Algorithm 2 and $k$-means for server-side clustering of the updated models.

---

**Algorithm 1** An algorithm for personalized federated learning with formal privacy guarantees in local neighborhoods.

---

**Require:** number of clusters $k$; initial hypotheses $\theta_j^{(0)}, j \in \{1, \ldots, k\}$; number of rounds $T$; number of users per round $U$; number of local epochs $E$; local step size $s$; user batch size $B_s$; noise multiplier $\nu$; local dataset $Z_c$ held by user $c$.

1: **for** $t = \{0, 1, \ldots, T-1\}$ **do**          ▷ Server-side loop
2:      $C^{(t)} \leftarrow \text{SampleUserSubset}(U)$
3:      $\text{BroadcastParameterVectors}(C^{(t)}; \theta_j^{(t)}, j \in \{1, \ldots, k\})$
4:      **for** $c \in C^{(t)}$ **do** in parallel          ▷ Client-side loop
5:          $\bar{j} = \arg\min_{j \in \{1, \ldots, k\}} F_c(\theta_j^{(t)}; Z_c)$
6:          $\theta_{\bar{j},c}^{(t)} \leftarrow \text{LocalUpdate}(\theta_{\bar{j}}^{(t)}; s; E; Z_c)$
7:          $\hat{\theta}_{\bar{j},c}^{(t)} \leftarrow \text{SanitizeUpdate}(\theta_{\bar{j},c}^{(t)}; \nu)$
8:      **end for**
9:      $\{S_1, \ldots, S_k\} = \text{k-means}(\hat{\theta}_{\bar{j},c}^{(t)}, c \in C^{(t)}; \theta_j^{(t)}, j \in \{1, \ldots, k\})$
10:     $\theta_j^{(t+1)} \leftarrow \frac{1}{|S_j|} \sum_{c \in S_j} \hat{\theta}_{\bar{j},c}^{(t)}, \quad \forall j \in \{1, \ldots, k\}$
11: **end for**

---

## 3.1 The Laplace mechanism under Euclidean distance in $\mathbb{R}^n$

---

**Algorithm 2** SanitizeUpdate obfuscates a vector $\theta \in \mathbb{R}^n$, with a Laplacian noise tuned on the radius of a certain neighborhood and centered in 0.

---

1: **function** SANITIZEUPDATE($\theta_{\bar{j}}^{(t)}; \theta_{\bar{j},c}^{(t)}; \nu$)
2:      $\delta_c^{(t)} = \theta_{\bar{j},c}^{(t)} - \theta_{\bar{j}}^{(t)}$
3:      $\varepsilon = \frac{n}{\nu \|\delta_c^{(t)}\|}$
4:      Sample $\rho \sim \mathcal{L}_{0,\varepsilon}(x)$
5:      $\hat{\theta}_{\bar{j},c}^{(t)} = \theta_{\bar{j},c}^{(t)} + \rho$
6:      **return** $\hat{\theta}_{\bar{j},c}^{(t)}$
7: **end function**

---

Algorithm 2's `SanitizeUpdate` is based on a generalization of the Laplace mechanism under Euclidean distance to $\mathbb{R}^n$, introduced in [4] for geo-indistinguishability in $\mathbb{R}^2$. The motivation to adopt the $L_2$ norm as distance measure is twofold. First, clustering is performed on $\theta$ with the $k$-means algorithm under Euclidean distance. Since we define clusters or groups of users based on how close their model parameters are under $L_2$ norm, we are looking for a $d$-privacy mechanism that obfuscates the reported values within a certain group and allows the server to discern among users belonging to different clusters. Second, parameters that are sanitized by equidistant noise vectors in $L_2$ norm are also equiprobable by construction and lead to the same bound in the increase of the cost function in first order approximation, as shown in Proposition 3.2. The Laplace mechanism under Euclidean distance in a generic space $\mathbb{R}^n$ is defined in Proposition 3.1.

**Proposition 3.1.** Let $\mathcal{L}_\varepsilon: \mathbb{R}^n \mapsto \mathbb{R}^n$ be the Laplace mechanism of the form $\mathcal{L}_{x_0,\varepsilon}(x) = \mathbb{P}\left[\mathcal{L}_\varepsilon(x_0) = x\right] = Ke^{-\varepsilon d(x,x_0)}$ with $d(.)$ being the Euclidean distance. If $\rho \sim \mathcal{L}_{x_0,\varepsilon}(x)$, then:

1. $\mathcal{L}_{x_0,\varepsilon}$ is $\varepsilon$-$d$-private and $K = \frac{\varepsilon^n \Gamma(\frac{n}{2})}{2\pi^{\frac{n}{2}} \Gamma(n)}$

2. $\|\rho\|_2 \sim \gamma_{\varepsilon,n}(r) = \frac{\varepsilon^n e^{-\varepsilon r} r^{n-1}}{\Gamma(n)}$

3. The $i^{th}$ component of $\rho$ has variance $\sigma_{\rho_i}^2 = \frac{n+1}{\varepsilon^2}$

where $\Gamma(n)$ is the Gamma function defined for positive reals as $\int_0^\infty t^{n-1} e^{-t}\, dt$ which reduces to the factorial function whenever $n \in \mathbb{N}$.

**Proposition 3.2.** Let $y = f(x, \theta)$ be the fitting function of a machine learning model parameterized by $\theta$, and $(X, Y) = Z$ the dataset over which the RMSE loss function $F(Z, \theta)$ is to be minimized, with $x \in X$ and $y \in Y$. If $\rho \sim \mathcal{L}_{0,\varepsilon}$, the bound on the increase of the cost function does not depend on the direction of $\rho$, in first order approximation, and:

$$\|F(Z, \theta + \rho)\|_2 - \|F(Z, \theta)\|_2 \leq \|J_f(X, \theta)\|_2 \|\rho\|_2 + o(\|J_f(X, \theta) \cdot \rho\|_2)$$

The results in Proposition 3.1 allow to reduce the problem of sampling a point from Laplace to i) sampling the norm of such point according to Equation (11) and then ii) sample uniformly a unit (directional) vector from the hypersphere in $\mathbb{R}^n$. Much like DP, $d$-privacy provides a means to compute the total privacy parameters in case of repeated queries, a result known as Compositionality Theorem for $d$-privacy 3.1. Although it was known as a folk result, we provide a formal proof.

**Theorem 3.1.** *Let $\mathcal{K}_i$ be $(\varepsilon_i)$-$d$-private mechanism for $i \in \{1, 2\}$. Then their independent composition is $(\varepsilon_1 + \varepsilon_2)$-$d$-private.*

**A heuristic for defining the neighborhood of a client**  At the $t^{\text{th}}$ iteration, when a user $c$ calls the `SanitizeUpdate` routine in Algorithm 2, it has already received a set of hypotheses, optimized $\theta_{\bar{j}}^{(t)}$ (the one that fits best its data distribution), and got $\theta_{\bar{j},c}^{(t)}$. It is reasonable to assume that clients whose datasets are sampled from the same underlying data distribution $\mathcal{D}_{\bar{j}}$ will perform an update similar to $\delta_c^{(t)}$. Therefore, we enforce points which are within the $\delta_c^{(t)}$-neighborhood of $\hat{\theta}_{\bar{j},c}^{(t)}$ to be indistinguishable. To provide this guarantee, we tune the Laplace mechanism such that the points within the neighborhood are $\varepsilon\|\delta_c^{(t)}\|_2$ differentially private. With the choice of $\varepsilon = n/(\nu\delta_c^{(t)})$, one finds that $\varepsilon\|\delta_c^{(t)}\|_2 = n/\nu$, and we call $\nu$ the *noise multiplier*. It is straightforward to observe that the larger the value of $\nu$ gets, the stronger is the privacy guarantee. This results from the norm of the noise vector sampled from the Laplace distribution being distributed according to Equation (11) whose expected value is $\mathbb{E}\left[\gamma_{\varepsilon,n}(r)\right] = n/\varepsilon$.

# 4  Experiments

**Synthetic data**  We generate data according to $k = 2$ different distributions: $y = x^T \theta_i^* + u$ and $u \sim \text{Uniform}[0, 1)$, $\forall i \in \{1, 2\}$ and $\theta_1^* = [+5, +6]^T$, $\theta_2^* = [+4, -4.5]^T$. We then assess how training progresses as we move from the Federated Averaging [19] (Figure 1a, 1b, 1c), to IFCA (Figure 1d, 1e, 1f), and finally Algorithm 1 (Figure 1g, 1h, 1i). Figure 2 provides the maximum value of privacy leakage clients incur into, per cluster. Further details about the experimental settings are provided in Appendix B.

**Hospital charge data**  This experiment is performed on the Hospital Charge Dataset by the Centers for Medicare and Medicaid Services of the US Government [1]. The healthcare providers are considered the set of clients willing to train a machine learning model with federated learning. The goal is predicting the cost of a service given where it is performed in the country, and what kind of procedure it is. More details on the preprocessing and training settings are included in Appendix B. To assess the trade-off between privacy, personalization and accuracy, a different number of initial hypotheses has been checked, as it is not known a-priori how many distributions generated the data. Accuracy has been evaluated at different levels of the noise multiplier $\nu$. Results are shown in Figure 4. Figure 3 provides the empirical privacy leakage distribution of the clients involved in a

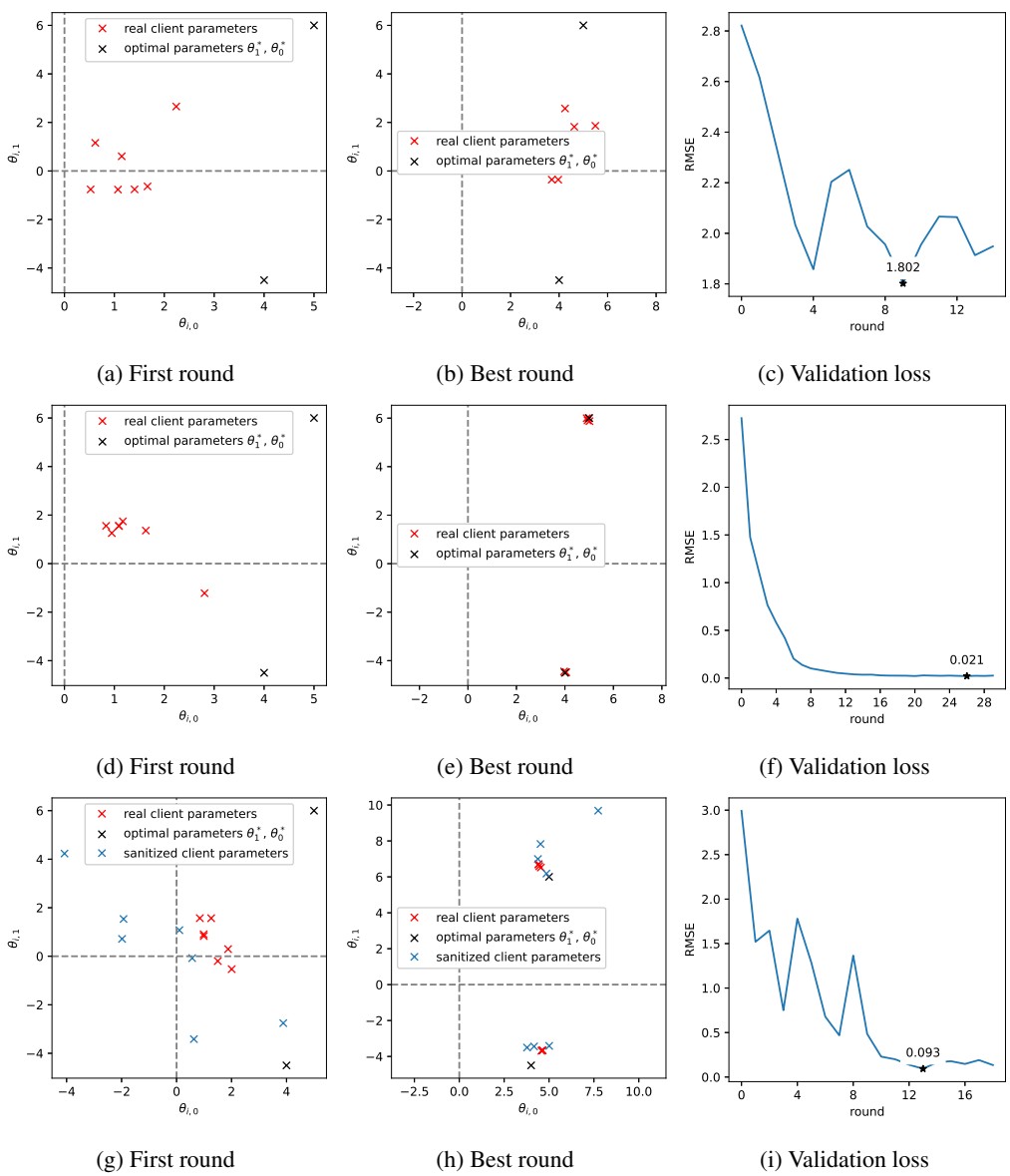

Figure 1: Learning federated linear models with: (a, b, c) one initial hypothesis and non-sanitized communication, (d, e, f) two initial hypotheses and non-sanitized communication, (g, h, i) two initial hypotheses and sanitized communication. The first two figures of each row show the parameter vectors released by the clients to the server.

| $\nu$ | Hypotheses | | | |
|---|---|---|---|---|
| | 7 | 5 | 3 | 1 |
| 0 | $-, -$ | $-, -$ | $-, -$ | $-, -$ |
| 0.1 | 517.0, 1551.0 | 418.0, 1342.0 | 473.0, 1386.0 | 528.0, 1540.0 |
| 1 | 36.3, 126.5 | 40.7, 127.6 | 44.0, 138.6 | 49.5, 147.4 |
| 2 | 15.4, 57.8 | 14.3, 54.5 | 22.0, 69.3 | 21.5, 66.6 |
| 3 | 7.7, 32.3 | 8.4, 36.7 | 12.5, 40.0 | 12.1, 40.0 |
| 5 | 5.7, 21.3 | 5.9, 22.0 | 5.5, 21.6 | 5.3, 20.9 |

Table 1: Hospital charge data: median and maximum local privacy budgets over the whole set of clients, averaged over 10 runs with different seeds. $\nu = 0$ means no privacy guarantee.

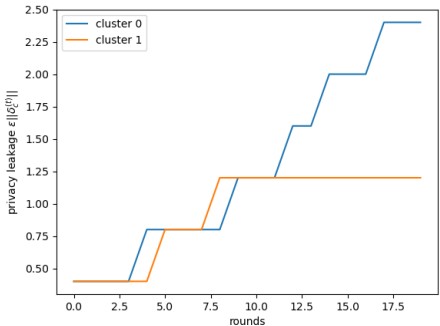

Figure 2: Synthetic data: max privacy leakage among clients clients. Privacy leakage is constant when clients with the largest privacy leakage are not sampled (by chance) to participate in those rounds.

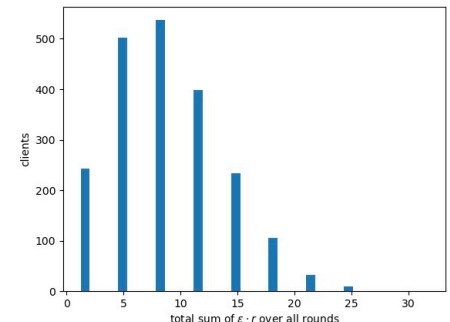

Figure 3: Hospital charge data: the empirical distribution of the privacy budget over the clients for: $\nu = 3$, 5 initial hypotheses, seed $= 3$, $r$ is the radius of the neighborhood, the total number of clients is 2062.

particular training configuration. Table 1 shows privacy leakage statics over multiple rounds and for all configurations.

**FEMNIST Image Classification [8]** Details on the experimental settings are in Appendix B. With the choice of the range of noise multipliers $\nu$ the corresponding value for the privacy leakage $\varepsilon \|\delta_c^{(t)}\|_2 = n/\nu$ would be enormous, considering a CNN with $n = 206590$ parameters, providing no meaningful theoretical privacy guarantees. This is a common issue for local privacy mechanisms [6],

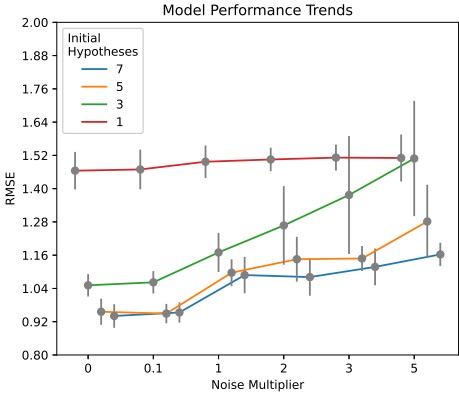

Figure 4: RMSE for models trained with Algorithm 1 on the Hospital Charge Dataset. Error bars show $\pm\sigma$, with $\sigma$ the empirical standard deviation. Lower RMSE values are better for accuracy.

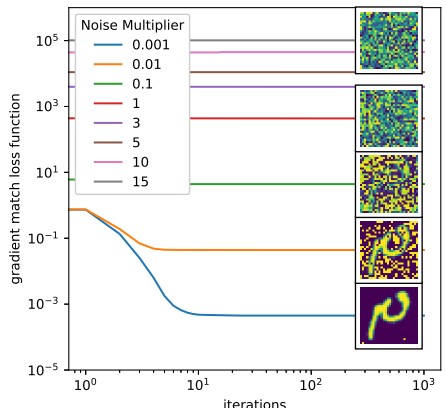

Figure 5: Effects of the Laplace mechanism in Proposition 3.1 with different noise multipliers as a defense strategy against the DLG attack.

|  | Cross Entropy loss | | RMSE loss | |
|---|---|---|---|---|
| $\nu$ | Average Accuracy | Standard Deviation | Average Accuracy | Standard Deviation |
| 0 | 0.832 | ± 0.012 | 0.801 | ± 0.001 |
| 0.001 | 0.843 | ± 0.006 | 0.813 | ± 0.014 |
| 0.01 | 0.832 | ± 0.017 | 0.805 | ± 0.008 |
| 0.1 | 0.834 | ± 0.026 | 0.808 | ± 0.019 |
| 1 | 0.834 | ± 0.014 | 0.814 | ± 0.012 |
| 3 | 0.835 | ± 0.017 | 0.825 | ± 0.010 |
| 5 | 0.812 | ± 0.016 | 0.787 | ± 0.003 |
| 10 | 0.692 | ± 0.002 | 0.687 | ± 0.014 |
| 15 | 0.561 | ± 0.005 | 0.622 | ± 0.003 |

Table 2: Effects of increasing the noise multiplier on the validation accuracy and standard deviation.

and it comes from the linear dependence on $n$: $\mathbb{E}\left[\gamma_{\varepsilon,n}(r)\right] = n/\varepsilon$. Still, it is possible to validate, in practice, whether this particular generalization of the Laplace mechanism can protect against a *specific* attack: DLG [31]. Figure 5 and Table 2 report the results of varying the noise multiplier values. When $\nu = 10^{-3}$ the ground truth image is fully reconstructed. Up to $\nu = 10^{-1}$ we see that at least partial reconstruction is possible. For $\nu \geq 1$ we see that, experimentally, the DLG attack fails to reconstruct input samples.

# 5    Conclusion

We use the framework of $d$-privacy to sanitize points in the parameter space of machine learning models, which are then communicated to a central server for aggregation in order to converge to the optimal parameters and, thus, obtain the personalized models for the diverse datasets. Given that the distribution of the data among individuals is unknown, it is reasonable to assume a mixture of multiple distributions. Clustering the sanitized parameter vectors released by the clients with the $k$-means algorithm shows to be a good proxy for aggregating clients with similar data distributions. This is possible because $d$-private mechanisms preserve the topology of the domain of true values. Our mechanism shows to be promising when machine learning models have a *small* number of parameters. Although formal privacy guarantees degrade sharply with large machine learning models, we show experimentally that the Laplace mechanism is effective against the DLG attack.

## Acknowledgments and Disclosure of Funding

The work of Sayan Biswas, Kangsoo Jung, and Catuscia Palamidessi was supported by the European Research Council (ERC) project HYPATIA under the European Unions Horizon 2020 research and innovation programme (grant agreement no. 835294).

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

# A Proofs

**Proposition 3.1.** Let $\mathcal{L}_\varepsilon \colon \mathbb{R}^n \mapsto \mathbb{R}^n$ be the Laplace mechanism of the form $\mathcal{L}_{x_0,\varepsilon}(x) = \mathbb{P}\left[\mathcal{L}_\varepsilon(x_0) = x\right] = Ke^{-\varepsilon d(x,x_0)}$ with $d(.)$ being the Euclidean distance. If $\rho \sim \mathcal{L}_{x_0,\varepsilon}(x)$, then:

1. $\mathcal{L}_{x_0,\varepsilon}$ is $\varepsilon$-$d$-private and $K = \frac{\varepsilon^n \Gamma(\frac{n}{2})}{2\pi^{\frac{n}{2}}\Gamma(n)}$

2. $\|\rho\|_2 \sim \gamma_{\varepsilon,n}(r) = \frac{\varepsilon^n e^{-\varepsilon r} r^{n-1}}{\Gamma(n)}$

3. The $i^{th}$ component of $\rho$ has variance $\sigma_{\rho_i}^2 = \frac{n+1}{\varepsilon^2}$

where $\Gamma(n)$ is the Gamma function defined for positive reals as $\int_0^\infty t^{n-1}e^{-t}\,dt$ which reduces to the factorial function whenever $n \in \mathbb{N}$.

*Proof.* We provide proofs for all three statements separately:

**1.** If $\mathcal{L}_{x_0,\varepsilon}(x) = Ke^{-\varepsilon d(x,x_0)}$ is a probability density function of a point $x \in \mathbb{R}^n$ then $K$ should be such that $\int_{\mathbb{R}^n} \mathcal{L}_{x_0}(x)dx = 1$. We note that it depends only on the distance $x$ and $x_0$ and we can write $Ke^{-\varepsilon d(x,x_0)}$ as $Ke^{-\varepsilon r}$ where $r$ is the radius of the ball in $\mathbb{R}^n$ centered in $x_0$. Without loss of generality, let us now take $x_0 = 0$. The probability density of the event $x \in \mathbb{S}_n(r) = \{x : \|x\|_2 = r\}$ is then $p(x \in \mathbb{S}_n(r)) = Ke^{-\varepsilon r}S_n(1)r^{n-1}$ where $S_n(1)$ is the surface of the unitary ball in $\mathbb{R}^n$ and $S_n(r) = S_n(1)r^{n-1}$ is the surface of a generic ball of radius $r$. Given that

$$S_n(1) = \frac{2\pi^{n/2}}{\Gamma(\frac{n}{2})} \tag{6}$$

solving

$$\int_0^{+\infty} \mathbb{P}\left[x \in \mathbb{S}_n(r)\right] dr = \int_0^{+\infty} Ke^{-\varepsilon r}S_n(1)r^{n-1}dr = $$
$$K\frac{2\pi^{n/2}\Gamma(n)}{\varepsilon^n \Gamma(\frac{n}{2})} = 1 \tag{7}$$

results in

$$K = \frac{\varepsilon^n \Gamma(\frac{n}{2})}{2\pi^{\frac{n}{2}}\Gamma(n)} \tag{8}$$

where $\Gamma(\cdot)$ denotes the gamma function. By plugging $\mathcal{L}_{x_0,\varepsilon}(x) = Ke^{-\varepsilon d(x,x_0)}$ in Equation 5:

$$Ke^{-\varepsilon d(x,x_1)} \leq e^{\varepsilon d(x_1,x_2)}Ke^{-\varepsilon d(x,x_2)} \tag{9}$$

$$e^{\varepsilon(\|x-x_2\|_2 - \|x-x_1\|_2)} \leq e^{\varepsilon\|x_1-x_2\|} = e^{\varepsilon d(x_1,x_2)} \tag{10}$$

which completes the poof of the first statement.

**2.** Without loss of generality, let us take $x_0 = 0$. Exploiting the radial symmetry of the Laplace distribution, we note that, in order to sample a point $\rho \sim \mathcal{L}_{x_0,\varepsilon}(x)$ in $\mathbb{R}^n$, it is possible to first sample the set of points distant $d(x,0) = r$ from $x_0$ and then sample uniformly from the resulting hypersphere. Accordingly, the p.d.f. of the $L_2$-norm of $\rho$ is the p.d.f. of the event $\rho \in \mathbb{S}_n(r) = \{\rho : \|\rho\|_2 = r\}$ which is then $\mathbb{P}\left[\rho \in \mathbb{S}_n(r)\right] = Ke^{-\varepsilon r}\mathbb{S}_n(1)r^{n-1}$, where $\mathbb{S}_n(r)$ is the surface of the sphere with radius $r$ in $\mathbb{R}^n$. Hence, we can write

$$\|\rho\|_2 \sim \gamma_{\varepsilon,n}(r) = \frac{\varepsilon^n e^{-\varepsilon r} r^{n-1}}{\Gamma(n)} \tag{11}$$

which completes the proof of the second statement.

**3.** With $\rho \sim \gamma_{\varepsilon,n}$ we have that, by construction,

$$\mathbb{E}\left[\rho^2\right] = \mathbb{E}\left[\sum_{i=1}^n \rho_i^2\right] = n\mathbb{E}\left[\rho_i^2\right] = n\sigma_{\rho_i}^2 \tag{12}$$

With the last equality holding since $\mathcal{L}_{0,\varepsilon}$ is isotropic and centered in zero. Recalling that

$$\mathbb{E}\left[\rho^2\right] = \frac{d^2}{dt^2}M_\rho(t)\bigg|_{t=0} \tag{13}$$

with $M_\rho(t)$ the moment generating function of the gamma distribution $\gamma_{\varepsilon,n}$,

$$\frac{d^2}{dt^2}\left(\left(1-\frac{t}{\varepsilon}\right)^{-n}\right)\bigg|_{t=0} =$$
$$= \frac{n(n+1)}{\varepsilon^2}\left(1-\frac{t}{\varepsilon}\right)^{-(n+2)}\bigg|_{t=0} =$$
$$= \frac{n(n+1)}{\varepsilon^2}$$

which leads to $\sigma_{\rho_i}^2 = \frac{n+1}{\varepsilon^2}$, completing the proof of the third statement and of the Proposition. □

**Proposition 3.2.** Let $y = f(x,\theta)$ be the fitting function of a machine learning model parameterized by $\theta$, and $(X,Y) = Z$ the dataset over which the RMSE loss function $F(Z,\theta)$ is to be minimized, with $x \in X$ and $y \in Y$. If $\rho \sim \mathcal{L}_{0,\varepsilon}$, the bound on the increase of the cost function does not depend on the direction of $\rho$, in first order approximation, and:

$$\|F(Z,\theta+\rho)\|_2 - \|F(Z,\theta)\|_2 \le \|J_f(X,\theta)\|_2\|\rho\|_2 + o(\|J_f(X,\theta)\cdot\rho\|_2)$$

*Proof.* The Root Mean Square Error loss function is defined as:

$$F = \sqrt{\frac{\sum_{i=1}^{|Z|}(f(x_i,\theta) - y_i)^2}{|Z|}} = \frac{\|f(X,\theta) - Y\|_2}{\sqrt{|Z|}} \tag{14}$$

If the model parameters $\theta$ are sanitized by the addition of a random vector $\rho \sim \mathcal{L}_{0,\varepsilon}$, we can evaluate how the cost function would change with respect to the non-sanitized parameters. Dropping the multiplicative constant we find:

$$\|f(X,\theta+\rho) - Y\|_2 - \|f(X,\theta) - Y\|_2 \le$$
$$\|f(X,\theta+\rho) - Y - f(X,\theta) + Y\|_2 =$$
$$\|f(X,\theta+\rho) - f(X,\theta)\|_2 =$$
$$\|f(X,\theta) + J_f(X,\theta)\cdot\rho - f(x,\theta) + o(J_f(X,\theta)\cdot\rho)\|_2 =$$
$$\|J_f(X,\theta)\cdot\rho + o(J_f(X,\theta)\cdot\rho)\|_2 \le$$
$$\|J_f(X,\theta)\|_2\|\rho\|_2 + o(\|J_f(X,\theta)\cdot\rho\|_2)$$

□

with $J_f(X,\theta)$ being the Jacobian of $f$ with respect to $X$ and $o(.)$ being higher terms coming from the Taylor expansion. Thus we proved that the bound on the increase of the cost function and that it does not depend on the direction of the additive noise, but on its norm, in first order approximation.

**Theorem 3.1.** *Let $\mathcal{K}_i$ be $(\varepsilon_i)$-d-private mechanism for $i \in \{1,2\}$. Then their independent composition is $(\varepsilon_1 + \varepsilon_2)$-d-private.*

*Proof.* Let us simplify the notation and denote:

$$P_i = \mathbb{P}_{\mathcal{K}_i}\left[y_i \in S_i | x_i\right]$$
$$P_i' = \mathbb{P}_{\mathcal{K}_i}\left[y_i \in S_i | x_i'\right]$$

for $i \in \{1,2\}$. As mechanisms $\mathcal{K}_1$ and $\mathcal{K}_2$ are applied independently, we have:

$$\mathbb{P}_{\mathcal{K}_1,\mathcal{K}_2}\left[(y_1,y_2) \in S_1 \times S_2 | (x_1,x_2)\right] = P_1.P_2$$
$$\mathbb{P}_{\mathcal{K}_1,\mathcal{K}_2}\left[(y_1,y_2) \in S_1 \times S_2 | (x_1',x_2')\right] = P_1'.P_2'$$

Therefore, we obtain:

$$\mathbb{P}_{\mathcal{K}_1,\mathcal{K}_2}\left[(y_1,y_2) \in S_1 \times S_2 | (x_1,x_2)\right] = P_1.P_2$$
$$\leq \left(e^{\varepsilon_1\,d(x_1,x_1')}P_1'\right)\left(e^{\varepsilon_2\,d(x_2,x_2')}P_2'\right)$$
$$\leq e^{\varepsilon_1\,d(x_1,x_1')+\varepsilon_2\,d(x_2,x_2')}\mathbb{P}_{\mathcal{K}_1,\mathcal{K}_2}\left[(y_1,y_2) \in S_1 \times S_2 | (x_1',x_2')\right]$$

$\square$

## B  Experimental settings

### B.1  Synthetic data

A total of 100 users holding 10 samples each, drawn from either one of the distributions, participate in a training of two initial hypotheses which are sampled from a Gaussian distribution centered in 0 and unit variance at iteration $t = 0$. A total of $U = 7$ users are asked to participate in the optimization at each round and train locally the hypothesis that fits better their dataset for $E = 1$ epochs each time. The noise multiplier is set to $\nu = 5$. Local step size $s = 0.1$ and a batch size $B_s = 10$ complete the required inputs to the algorithm. To verify the training process, another set of users with the same characteristics is held out form training to perform validation and stop the federated optimization once the is no improvement in the loss function in Equation (14) for 6 consecutive rounds. Although at first the updates seem to be distributed all over the domain, in just a few rounds of training the process converges to values very close to the two optimal parameters. With the heuristic presented in Section 3.1 it is easy to find that whenever a user participates in an optimization round it incurs in a privacy leakage of at most $n/\nu = 2/5 = 0.4$, in a differential private sense, with respect to points in its neighborhood. Using the result in Theorem 3.1 clients can compute the overall privacy leakage of the optimization process, should they be required to participate multiple times. For any user, whether to participate or not in a training round can be decided right before releasing the updated parameters, in case that would increase the privacy leakage above a threshold value decided beforehand.

### B.2  Hospital charge data

The dataset contains details about charges for the 100 most common inpatient services and the 30 most common outpatient services. It shows a great variety of charges applied by healthcare providers with details mostly related to the type of service and the location of the provider. Preprocessing of the dataset includes a number of procedures, the most important of which are described here:

i) Selection of the 4 most widely treated conditions, which amount to simple pneumonia; kidney and urinary tract infections; hart failure and shock; esophagitis and digestive system disorders.

ii) Transformation of ZIP codes into numerical coordinates in terms of longitude and latitude.

iii) Setting as target the Average Total Payments, i.e. the cost of the service averaged among the times it was given by a certain provider.

iv) As it is a standard procedure in the context of gradient-based optimization, dependent and independent variables are brought to be in the range of the *units* before being fed to the machine learning model. Note that this point takes the spot of the common feature normalization and standardization procedures, which we decided not to perform here to keep the setting as realistic as possible. In fact, both would require the knowledge of the empirical distribution of all the data. Although it is available in simulation, it would not be available in a real scenario, as each user would only have access to their dataset.

Given the preprocessing described above, the dataset results in 2947 clients, randomly split in train and validation subsets with 70 and 30 per cent of the total clients each. The goal is being able to predict the cost that a service would require given where it is performed in the country, and what kind of procedure it is. The model that was adopted in this context is a fully connected neural network (NN) of two layers, with a total of 11 parameters and Rectified Linear Unit (ReLU) activation function. Inputs to the model are an increasing index which uniquely defines the healthcare service, the longitude and latitude of the provider. Output of the model is the expected cost. Tests have

been performed to minimize the RMSE loss on the clients selected for training (100 per round) and at each round the performance of the model is checked against a held-out set of validation clients, from where 200 are sampled every time. If 30 validation rounds are passed without improvement in the cost function, the optimization process is terminated. In order to decrease variability of the results, a total of 10 runs have been performed with different seeds for every combination of number of hypotheses and noise multiplier.

## B.3 FEMNIST image classification

The task consists in performing image classification on the FEMNIST [8] dataset, which is a standard benchmark dataset for federated learning, based on EMNIST [10] and with the data points grouped by user. It consists of a large number of images of handwritten digits, lower and upper case letters of the Latin alphabet. As a preprocessing step, images of client $c$ are rotated 90 degrees counter-clockwise depending on the realization of the random variable $rot_c \sim \text{Bernoulli}(0.5)$. This is a common practice in machine learning to simulate local datasets held by different clients being generated by different distributions [14, 15, 18, 21]. The chosen architecture is described in Table 3 and yields a parameter vector $\theta \in \mathbb{R}^{n_0}$, $n_0 = 1206590$. Runs are performed with a maximum of 500 rounds of federated optimization, unless 5 consecutive validation rounds are conducted without improvements on the validation loss. The latter is evaluated on a held out set of clients, consisting of 10% of the total number. Validation is performed every 5 training rounds, thus the process terminates after 25 rounds without model's performance improvement. The optimization process aims to minimize either the RMSE loss or the Cross Entropy loss [29] between model's predictions and the target class.

| Layer | Properties |
|---|---|
| 2D Convolution | kernel size: (2,2) 
 stride: (1,1) 
 nonlinearity: ReLU 
 output features: 32 |
| 2D Convolution | kernel size: (2,2) 
 stride: (1,1) 
 nonlinearity: ReLU 
 output features: 64 |
| 2D Max Pool | kernel size: (2,2) 
 stride: (2,2) 
 nonlinearity: ReLU |
| Fully Connected | nonlinearity: ReLU 
 units: 128 |
| Fully Connected | nonlinearity: ReLU 
 units: 62 |

Table 3: NN architecture adopted in the experiments of Section 4

