# OpenReview forum: "Group privacy for personalized federated learning"
_NeurIPS.cc/2022/Workshop/Federated_Learning — FL-NeurIPS 2022 Oral_

### Official Review · Reviewer_c2pp · 2022-10-17
**Solid paper, definite accept**

This paper advocates for the use of metric DP in personalized federated learning, in particular, in the context of the clustered personalized FL framework introduced in Ghosh et al. (2020). The idea is reasonably straightforward, with clients applying a SanitizeUpdate function to their updates (generalizing the well-known Laplace mechanism) before sending them to the server. One key aspect is that adopting metric DP enables the sanitization to be applied directly in model parameter space rather than to model-updates, as is more commonly done in non-personalized FL.

The work is interesting and well-done and should be of interest to the FL-NeurIPS community. Thus I recommend it be accepted.

Note that the use of metric DP in the context of Federated Learning has previously been discussed, albeit in a very different context (in terms of jointly compressing and sanitizing updates) in [Chaudhuri et al. (2022)](https://arxiv.org/abs/2203.08134).

Since metric DP provides a different notion of privacy compared to other forms of DP that have been more widely considered in the FL literature, it could be informative to add a comparison of the different approaches in terms of their effectiveness against data reconstruction attacks.

Minor:
* Define or describe the $\Gamma$ function used in Prop 3.1

---

### Official Review · Reviewer_kWjA · 2022-10-17
**Intuitive use of metric local DP on mixture of distributons in FL setting.**

The authors consider a heterogeneous distribution of groups of users having similar (if not homogeneous) data distribution for Federated Learning (FL); "personalized FL" in this setting is to learn models best suited for each group.

The authors use metric based local differential privacy (DP) to preserve the privacy of individual client updates and subsequently use *k*-means on the model updates from clients to obtain a set of model updates for each group of users, which are then aggregated to yield one model per group on the server. The key assumption that the authors make is that clients with similar data distribution will also have similar model updates.

**Strengths:**

The proposal is indeed simple, clear, and intuitive. The authors validate their assumptions on synthetic dataset and also present experimental results on a real-world dataset as well as on FEMNIST task. The use of metric-DP for a generic machine learning task in FL setting to address data heterogeneity seems to be an original contribution.

**Weaknesses:**
1. The experiments suggest that this approach is useful for small models (with few parameters), but breaks down significantly for larger models; largely due to local-DP.
2. Furthermore, it is not clear to me if the heuristic used to define the client’s neighbourhood, which also forms the basis for using metric-DP, are valid for all scenarios.
3. On a more practical side, figuring out the “k” in k-means is a challenging issue to solve in a real world scenario.
4. When a large value of ‘k’ is impractical in a real world setting due to communication cost, it is unclear to me as to how well much worse the baseline FL setup (k=1) would be compared to k=n, where n is small.

**Conclusion**
The above weaknesses not with standing, I believe this paper proposes in an interesting direction and deserves more discussion.

---

### Official Review · Reviewer_TPaQ · 2022-10-18
**Review of paper54**

This paper considers two important problems in federated learning simultaneously, personalization of global model and leakage of private information from the client-server communication. This paper utilizes laplace mechanism to provide differential privacy on top of clustering algorithm, which aims to provide personalization. This paper also provide a formal proof of d-privacy of the proposed scheme, and emprically demonstrates convergence performance. However, the reviwer has the following major concerns on this paper:
1. One of major concerns is contribution and novelty of this paper. First, it is not new to utilize clustering algorithm in personalized federated learning as well as laplace mechanism to provide differential privacy. In addition, formal proof for d-privacy is also not beyond the standard proof techniques.
2. It does not provide convergence proof of the proposed algorithms.
3. As formal privacy guaratees degrade sharply with large machine learning models, practicality of the proposed algorithm is limited.

---

### Decision · Program_Chairs · 2022-10-20

Accept (Oral)